# TiO_2_ Nanoparticle Filler-Based Mixed-Matrix PES/CA Nanofiltration Membranes for Enhanced Desalination

**DOI:** 10.3390/membranes11060433

**Published:** 2021-06-09

**Authors:** Mehwish Batool, Amir Shafeeq, Bilal Haider, Nasir M. Ahmad

**Affiliations:** 1Institute of Chemical Engineering and Technology, University of the Punjab, Lahore 54590, Pakistan; mehwishbatool@cuilahore.edu.pk (M.B.); amirengr@gmail.com (A.S.); 2Department of Chemical Engineering, COMSATS University Islamabad, Lahore Campus, Lahore 54000, Pakistan; 3Polymer Research Lab, School of Chemical and Materials Engineering (SCME), National University of Sciences and Technology, H-12, Islamabad 44000, Pakistan

**Keywords:** titanium dioxide nanoparticles (TiO_2_ NPs), cellulose acetate (CA), Polyethersulfone (PES), Polyvinylpyrrolidone (PVP), mixed matrix nanocomposite membranes (MMNMs), desalination

## Abstract

Mixed-matrix nanocomposite (PES/CA/PVP) membranes were fabricated for water desalination by incorporating varying amount of titanium dioxide nanoparticles (TiO_2_ NPs) ranging from 0 and 2 wt. %. Efficient dispersion of nanoparticles within polymeric membranes was achieved using the chemical precipitation method for uniform surface generation, and an asymmetric morphology was achieved via phase inversion method. Finally, membranes were characterized by Fourier Transform Infrared (FTIR) spectroscopy, Thermo Gravimetric Analysis (TGA), Scanning Electron Microscopy (SEM), porosity and contact angle analysis. FTIR confirmed chemical composition of membranes in terms of polymers (PES/CA/PVP) and TiO_2_. TGA analysis confirmed an increase in thermal stability of membranes with the increase of TiO_2_ nanoparticles loading. The addition of TiO_2_ nanoparticles also resulted in an increase in porous structures due to an increase in mean pore size, as shown by SEM results. An increase in the hydrophilicity of the membranes was observed by increasing the concentration of TiO_2_ nanoparticles. The present study investigated pristine and mixed-matrix nanocomposite NF membrane performance while filtering a NaCl salt solution at varying concentration range (from 1 to 4 g/Lit 6 bar). The prepared membranes demonstrated significant improvement in water permeability and hydrophilicity. Further, to optimize the water flux and salt rejection, the concentration of Polyvinylpyrrolidone (PVP) was optimized along with TiO_2_ nanoparticles. Both the water flux and salt rejection of the fabricated membranes were observed to increase with an increase inTiO_2_ nanoparticles to 2 wt. % loading with optimized PVP concentration, which demonstrated the improved desalination performance of resultant membranes.

## 1. Introduction

Membrane separation technology for the desalination of brackish and seawater has been widely used for many years [1]. Efficient desalination of brackish water has been achieved by reverse osmosis and nanofiltration processes [2]. Conventional NF and RO membranes use a selective ultrathin barrier layer backed by multiple strong, porous polymer support layers. The thin film of aromatic polyamide coated on an asymmetric Polysulfone or polyethersulfone integrally skinned membrane casted by the phase inversion process comprise a traditional NF/RO membrane model [1]. The efficiency parameters of the nanofiltration process i.e., permeate flux and rejection factor. These are defined by membrane physical and chemical properties process variables, solvent/solute ratio, operating pressure, coagulant bath temperature, organic and inorganic additives. To address the issues of conventional polymeric membranes, nanocomposite membranes have gained significant attention for water purification over the last three decades [2,3,4,5].

To achieve a desired membrane morphology and efficiency, care must be taken to monitor the phase inversion process. Compositions of polymer solutions and coagulation media are important factors that affect the phase inversion mechanism during membrane synthesis process. To obtain hydrophilicity of membranes and control the phase separation process, pore former and filler are added to the casting the solution [6]. Polyethersulfone (PES) has sulfone group and ether bonds that alternate between aromatic rings and provide the overall degree of molecular surface modification, producing creep resistance, high stiffness, strength, and dimensional stability. The modification of PES membranes by blending suitable polymer may help to improve aforementioned characteristic. A cellulose acetate phthalate hydrophilic polymer blended with PES can increase the hydrophilicity of prepared membranes. Mohan et al. studied membranes of cellulose acetate/Polysulfone combined with polyethersulfone and used cellulose acetate as the base polymer [7]. The combination of CA and PES has also been stated in the literature to have good desalination properties, with a 99% salt rejection and a moderate flux of 21 kg/h·m^2^ [8]. Therefore, the blending of engineering thermoplastics such as Polysulfone or polyethersulfone with cellulose acetate is recommended, owing to their ability to withstand superior mechanical and chemical properties [9]. Nanofiltration membrane performance is found to be very effected by blending inorganic nanoparticles [10,11,12]. Nanoparticle incorporation into polymeric membranes such as titanium dioxide (TiO_2_) [13,14,15,16,17,18,19], graphene oxide(GO) [20,21,22,23,24], alumina (Al_2_O_3_) [25,26], iron oxide (Fe_3_O_4_) [27,28],carbon nanotube (CNTs) [29,30,31,32,33,34], Zeolitic imidazolate frameworks-8(ZIF 8) [35], copper oxide (Cu_2_O), zinc oxide (ZnO) [36], and activated carbon(AC) [37,38,39,40,41,42] have shown potential to improve strength, hydrophilicity and enhance thermo mechanical properties. It also improves filtration efficiency, including higher flux, permeability, salt rejection, and membrane life. The presence of an inorganic phase in polymer matrix can affect polymer chain molecular movement and increase the free volume between them. The mixed-matrix nanocomposite membrane is a potential and emerging technology for future applications. Titanium dioxide NPs have potential to remove salts from water, improve antibacterial activity, and for dye degradation. TiO_2_ NPs also have excellent hydrophilicity, thermal, permeability, and chemical stability. It makes TiO_2_ to be ideal filler for synthesizing these novel membranes. TiO_2_ loaded membranes have better properties compared to other NPs without compromising performance in terms of salt rejection and water flux. Similar to other fillers, titanium dioxide nanoparticles are being employed to membranes in order to increase salt removal from water [43,44].

This work focuses on improving nanofiltration membrane performance for desalination applications in terms of salt rejection and water flux. Polyethersulfone and cellulose acetate were taken as the principal polymers for a membrane synthesis with a PVP as pore former. PVP may work as a carrier for hydrophilic TiO_2_ nanoparticles during membrane fabrication, improving their compatibility with the polymer matrix. As a result, nanoparticle agglomeration can be minimized, and filler distribution can be improved. The resulting nanofiltration membranes have higher flux, as the structure of the membrane is such that a very thin skin layer is at the top. It is evident from the literature that studies on the effect of PVP and filler concentration on mixed-matrix nanocomposite membranes is a promising approach. Hence in this study nanofiltration membranes prepared via the phase inversion method and investigation of the effects of the PVP (pore former) and filler concentration on membrane morphology and performance are being reported. This proposed combination of polymers, pore former and filler is novel in its nature, with the aim of improving water flux due to presence of TiO_2_ nanoparticles in the range of 0.5 to 2 wt. %.

## 2. Materials and Methods

### 2.1. Materials

The following polymeric materials were used in this study. Cellulose acetate (CA) (CH_3_CO)_2_O (Mw = 102.09 g/mol) was purchased from Acros Organics (Geel, Belgium). Polyethersulfone (M_W_ = 75,000 g/mol, Tg = 225 °C) Ultra son^®^ E 6020 was supplied by BASF (Ludwigshafen, Germany). Polyvinylpyrrolidone (PVP) (average M_W_ = 40,000) in powder form (C_6_H_9_NO)_n_) was purchased from Sigma–Aldrich (St. Louis, MO, USA) as a pore former and hydrophilic agent. Titanium(IV) TiO_2_ oxide, Antase 8613-1405 (M_W_ = 79.88) and1–methyl–2–pyrrolidone (NMP) with 99.5% analytical purity as a non–solvent was purchased from Daejung Chemicals (Siheung-si, Korea). Sodium chloride (NaCl) was purchased from Chemsupply (Lahore, Pakistan).

### 2.2. Nanocomposite Mixed Matrix Membrane Prepration

The phase inversion method was used to fabricate mixed-matrix nanocomposite membranes (MMNMs). The total polymer concentration was maintained at 17.5 wt. %, which was achieved by dissolving the PES polymer cellulose acetate and PVP pore former in NMP as a solvent and vigorously stirring for a period of 3–4 h at 65 °C to achieve a nicely dispersed solution. Subsequently, titanium dioxide (TiO_2_) nanoparticles were mixed with the solvent. The mixture was stirred overnight and then added to the polymer-doped solution. After constant mechanical stirring, the mixed-matrix membranes were casted by a solution casting machine (Automatic Film Applicator, Elcometer4340 M43 6BU (Elcometer Limited, Manchester, UK) at a speed of 50–60 mm/s at room temperature by using water as anon–solvent via the phase inversion method in a coagulation bath for 25–30 s. After the phase inversion process, the resulting fabricated membranes were immersed in a water and isopropanol mixture (70/30) for 19 h then kept in glycerol for 4–5 h to preserve the pores and membrane structure. Finally, the fabricated membranes were dried at room temperature. The membrane thickness was maintained between 0.18–0.22 ± 0.02 mm. Figure 1 shows the steps for nanocomposite membrane synthesis using phase inversion process. During casting, the temperature was maintained at 18 ± 1 °C. Table 1 shows the polymer/pore former concentration and nanoparticles amount in the composite membranes.

In first instance, the PVP concentration was optimized to obtain a membrane with maximum water flux and salt rejection. Afterwards, the effect of TiO_2_ NP concentrations on salt rejection and water flux was also investigated as shown in Table 1. During the studies of PVP concentration optimization, the amount of NPs was also varied from 0.5 to 2 wt. %. This approach was followed in anticipation of effect of better distribution of TiO_2_ NPs in casting solution. Increased casting solution viscosity at high filler concentrations slowed the phase inversion process and allowed nanoparticles to collect together due to delayed demixing. The membranes were dried under ambient conditions and saved for later use. To mount membranes in the dead–end stirred permeation cell, the fabricated membranes were made to desired sizes.

### 2.3. Membrane Characterization

Composition of membranes and interaction between polymer chains of the related chemical bonds with the TiO_2_ and pore former were investigated. FTIR analysis was conducted using the Thermo-Nicolet 6700 P FTIR Spectrometer (Thermo Fisher Scientific, Madison, WI, USA). In the range of 4000–650 cm^−1^, FTIR spectra of 128 scans of pure and MMNMs were obtained.

SEM was used to analyze the membrane morphology and surface–structure of CA/PES/PVP mixed–matrix nanocomposite membranes with various percentages of nanoparticles. The images were obtained by scanning electron microscopy (Inspect S–50, Thermo Fisher Scientific, Madison, WI, USA) at 20 KV. The flat–sheet membrane specimens were cut into pieces of 0.25 cm^2^, immersed in liquid nitrogen, carefully fractured cryogenically for the cross–sectional view. The specimens were mounted on blocks, and then coated membrane samples via gold sputtering before analysis.

The contact angle of the membrane surface was analyzed by the sessile drop method to study the hydrophilicity of the fabricated membranes and the impact of the filler/pore former. On the top surface, 1 μL of water was carefully dropped, and the dynamic contact angle was determined using the optimal video analysis method. The figures reported are the average contact angle of deionized water droplets on each sample at three different locations.

Thermo gravimetric analysis was performed in which thermal changes in materials a function of temperature with a constant heating rate of weight losses and the thermal degradation of CA/PES and CA/PES with the nanoparticle loading (0–2 wt. %) composite membranes were studied by using the TGA701 system (LECO Inc., St. Joseph, MI, USA). The study of all samples under air at a 20 °C/min heating rate and a 35–800 °C temperature was carried out by using 2 ± 0.01 mg of samples.

The membranes were cut into circular shape with a diameter of 5 cm to fit into the dead–end stirred cell, and a magnetic stirrer was used to stir the solution (Spectrum, NF Cell-S76–400 Model, Gardena, CA, USA). To evaluate effect of pressure, a high purity N_2_ gas was used for which pressure was adjustable between 0 and 150 bar. The effective membrane area was 0.0020 m^2^, and nanofiltration experiments were conducted with feed solution up to 260 mL. The experiments were performed at room temperature (25 ± 2 °C) and a fixed pressure of up to 6 bar, and the membranes were soaked in ethanol for up to 10 min prior to the experiment. At least three readings per sample were obtained. The average of three readings was calculated and being reported. The salt rejection and permeate flux were also measured [45,46]. By using Equation (1) water flux J_w_ (L/m^2^h) was determined.
(1)$$Jw=\frac{Q}{A\cdot \Delta t}$$
where ∆*t* (hours) is the time interval, *Q* (Liter) is the permeate volume and *A* (m^2^) is the area of the membrane. The dead–end stirred cell had a fixed area of the membrane of *A* = 0.0020 m^2.^

Salt rejection efficiency *(%SR)* was calculated using Equation (2) [47].
(2)$$\%SR=\frac{Conductivity\;of\;permeate\;\left(Cp\right)}{Conductivity\;of\;feed\;\left(Cf\right)}\times 100$$

Using a conductivity meter (Cyber scan PC 300 Series, Lahore Pakistan), the conductivity of the feed and permeate solutions was measured. Distilled water was used for preparing feed solution.

The difference in weight between dry and wet membranes was used to measure water uptake. For this purpose, the prepared membrane was dipped in distilled water and cut into 4 cm × 4 cm pieces for 72 h. Water uptake was determined using Equation (3) [48].
(3)$$Water\;uptake=\frac{wet\;weight-dry\;weight}{wet\;weight}\times 100$$

By using Equation (4), the overall porosity (ε) of the asymmetric mixed-matrix membranes was calculated [49].
(4)$$ԑ=\frac{Wwet-Wdry}{A.l.\rho }$$
where *W_wet_* and *W_dry are_* the wet membrane weight and the dry membrane weight, respectively. ρ (998 kg/m^3^) is the water density, and *l* (m) is the thickness of the membrane.

## 3. Results and Discussions

### 3.1. Fourier Transform Infrared Spectroscopy Analysis

FTIR spectra of the fabricated membranes are shown in Figure 2. M4 is a pristine membrane of CA and PES without any filler. The broad peak at ~1461 cm^−1^ corresponded to the CH_3_–C–CH_3_ group of PES asymmetric stretching [50]. The peak at ~717 cm^−1^ represents the C=H bending vibrations of PES styrene groups [51]. A wavelength of 919 cm^−1^ relates to the C–O–H linkage of CA [23]. M4T1–M4T4 is a composite membrane of CA, PES with TiO_2_, and 7.5 wt. % PVP. Ass light peak shift from 3287 to 3351 cm^−1^ may be due to the addition of the PVP pore former was observed [52]. A short and weak peak at 1746 cm^−1^ is attributed to the C–H bending of the aromatic ring of CA [17,53]. In M4T2,the broad PVP peak FTIR spectrum at 3349 cm^−1^ represents the stretching vibration of the secondary amine N–H group present in PVP [54]. The peaks of 2936 and 2832 cm^−1^ were known as the stretching of asymmetric C–H alkanes [55] and PVP stretching of O–H, respectively. In the FTIR region of the PVP-based filler membrane, low-intensity bending of CH_2_ occurred from1151 to 1105 cm^−1^ [56].The stretching vibrations of symmetric S = O and asymmetric S = O, respectively, can therefore be traced to the bands at 1151 and 1241 cm^−1^, which are attributed to PES [22]. The small peaks at 636 to 750 cm^−1^ illustrate the presence of TiO_2_ in the composite membrane [57]. M4T3 is a composite membrane of CA and PES with 1.5% TiO_2_, and 7.5 wt. % PVP [52]. The O–H stretching frequency is related to the band at 3304 cm^−1^ which confirms the CA sample as partially acetylated cellulose [54]. M4T4 is a composite membrane of CA, PES with 2% TiO_2_, and 7.5 wt. % PVP [17]. This peak increased in its intensity due to the relative increase in the concentration of PVP in the composite membranes. The sharp peak of 1784 cm^−1^ is the peak of the aromatic ring of PVP. This peak is so strong that it supports the overwhelming effect of PVP on the other organic precursor present in the composite membrane. The presence of the hydroxyl group can be attributed to the peak at 3287 cm^−1^ and the stretching of the C–O group, ether group, and C–O bond of the CH_2_–OH group peaks at 1642, 1374 and 1040 cm^−1^, respectively [58,59]. However, these peaks slightly shifted to 1029, 1733, 1224, and 3349 cm^−1^ with high intensity in the case of the CA–PES membrane [60]. The peaks at 750 and 636 cm^−1^ illustrate the presence of TiO_2_ in the composite membrane [61]. These two peaks are present because of the increased concentration of TiO_2_ as filler in the composite membrane.

### 3.2. Scanning Electron Microscopy Study

Figure 3 shows the effect of TiO_2_ nanoparticle concentration on membrane morphology at two different resolutions, i.e., 500 and 2000. All the membranes exhibit thick dense top layers and a finger-like morphology in the bottom layer, which confirms the asymmetric structure [7]. Porosity is important for the hydrophilic behavior of the membrane. The membrane is quite porous even without the addition of filler. With the addition of TiO_2_ (0.5 wt. %), the width of the finger-like pores increases with the formation of circular and irregular-shaped pores. A further increase in filler concentration results in an increase in void formation [62]. The increase in TiO_2_ concentration up to 2 wt. % results in enhanced macro void formation as well as higher structural porosity. It can be summarized that increases in the filler loading continue to increase the porosity of the membrane, which can also be confirmed by porosity analysis [63].

It can be summarized that in the casting solution, adding a small amount of TiO_2_ nanoparticles may lead to the formation of a denser membrane [64]. However, at a 2 wt. % NP concentration at a constant coagulation bath temperature (CBT = ±18 °C) resulted in more macro voids and porous structures, as shown in the SEM results.

### 3.3. Thermogravimetric Analysis

Thermal analysis of the pure and MMMs was performed. Figure 4 shows the TGA results for the pristine membrane and the TiO_2_ NP composite membrane.

The thermal stability of the pristine membrane and nanocomposite membrane was measured with a thermo gravimetric analyzer. It was observed that after incorporating NPs into the polymeric solution, the decomposition temperature (Td) increased and thermal stability was improved as compared to the pristine membrane. The composite membranes decomposed in three stages. The range of weight loss for M4 can be observed from 35 to 270 °C, which may be attributed to the loss of moisture due to volatile compounds. The TGA graph for the pristine membrane, however, indicates significant decomposition below 200 °C due to the loss of functional groups in the sample [19]. Secondly, the main thermal degradation and deacetylation process of the polymer matrix chain M4 can be observed from 270 to 420 °C. Finally, a decline at first leads to a constant thermal profile ranging from 420 to 650 °C, mainly due to the carbonization of the degraded product to ash. The addition of filler and pore former in the pristine membrane makes it a highly thermos table composite polymeric membrane (M4T1), of which the temperature of 5%weight loss is above 450 °C, and there is only one weight loss step attributed to the decomposition of the main polymer chain. Three weight loss transitions in three different temperature ranges can be distinguished in Figure 4 for PES/CA. The first step is due to the loss of absorbed water. The decomposition of the sulfone groups can be attributed to the second one, between 450 and 500 °C. The degradation of the main polymer chain is due to the third thermal degradation of CA at about 600 °C [65]. TGA further evaluates the thermal stability of composite membranes. For the M4T2 and M4T3 membranes, weight loss of around twenty percent was observed between room temperature and 200 °C because of surface evaporation and attached water molecules. In the temperature range of 300–450 °C, about a 30% second weight loss was observed due to the sulfonic group loss by desulfonation. At temperatures >450 °C, the breakage of the PES monomer main chain was the third weight loss regime for all membranes [66]. Compared to pure CA/PES, the composite membranes showed a significant change in thermal degradation temperature. Above 700 °C, an approximately constant mass was found, confirming the thermal stability of samples. It was observed that the incorporation of NPs into the polymeric membrane results in an improvement in the membrane thermal stability and decomposition temperature (Td) compared to the pure polymeric membrane. By comparing the TGA results, it was also confirmed that by increasing the filler concentration, the decomposition temperature continues to increase [64,67].

### 3.4. Percentage Porosity and Contact Angle Measurement

The overall porosity and contact angles of the mixed-matrix nanocomposite membranes are presented in Table 2 and Figure 5 and Figure 6. The porosity measurement results showed that all prepared M4T1–M4T4 composite membranes had high porosity in the 70–76% range compared to 52% for the pure CA/PES (M4) membrane prepared without any filler (Table 2). Figure 5 shows that with an increase in TiO_2_ concentration, the percentage porosity of the nanocomposite membranes increases. This means that the porosity of the internal structure of the membrane increases similar to TiO_2_, which also enhances the membrane’s properties and can contribute to improved directional flow rates through the membranes [64].

To determine the hydrophilicity, the contact angles of the membranes were measured. This is an indirect evaluation and a major parameter. By using the sessile drop technique, the hydrophilicity of the M4–M4T4 composite membranes was calculated.

The contact angles were calculated five times and the averages for synthesized membranes are compiled in Table 2. The highest contact angle was measured for M4, which showed the lowest hydrophilic characteristics of the membrane. Compared to bare membranes, a lower contact angle was detected in the composite membrane. Moreover, the contact angle decreased from 57.7 to 21° as the percentage of TiO_2_ increased from 1–2 wt. % due to an increase in hydrophilicity [68]. An important property to achieve a higher flux and permeation rate of nanocomposite membranes that can affect flux is surface hydrophilicity [69]. The lowest contact angle indicates that the surface of the membrane is more hydrophilic. With a water contact angle of 57.7 ± 3.2°, the pristine M4 membrane had the highest water contact angle. Increasing the amount of TiO_2_ by more than 2 wt. % did not enhance the hydrophilicity due to the aggregation of TiO_2_ NPs [67]. Increasing the amount of TiO_2_ by more than 2 wt. % did not enhance the hydrophilicity due to the aggregation of TiO_2_ NPs [67].

### 3.5. Water Uptake Analysis

Water uptake was directly related to the hydrophilicity of the membrane [64]. By using Equation (3), water uptake was calculated. The water uptake of the pure membrane was 56.83%, and after adding TiO_2_ up to 2 wt. %, it was 75.34%, as shown in Figure 7 and Table 2. The water uptake was observed to increase as TiO_2_ nanoparticles increased, Further, the PVP pore former concentration may have led to increased water uptake. The water uptake of the M4 to M4T4 membranes is shown in Table 2 [70]. TiO_2_ nanoparticles create spaces in the polymer matrix, which can increase the water uptake.

### 3.6. Water Flux

Water flux (WF) and salt rejection for the NF membranes are shown in Figure 8 and Figure 9. For desalination applications, nanofiltration membrane performance improved in terms of water flux and salt rejection. The WF is 65–89.6 L/m^2^h in a solution of 1000–4000 ppm NaCl and was used to realize salt rejection. The experiment was conducted at room temperature (20–25 °C), and the salt rejection of the membranes was calculated together with the permeate flux. It is observed that salt rejection is function of pressure, as pressure increases salt rejection also increases.

Figure 8 and Figure 9 show the average values of water flux for three measurements and salt rejection of the NaCl solution at various concentrations ranging from 1000 to 4000 ppm for the pure membrane as well as nanocomposite membranes, along with the impact of different TiO_2_ concentrations (0–2 wt. %.). In case of TiO_2_ blend membranes (2 wt. %) as compared to pure membranes, the water flux increased. The addition of titanium dioxide changed the surface porosity and enhanced the bulk properties, including skin layer thickness, porosity, and membrane thickness. Overall porosity in the range of 52 to 76% of the prepared nanofiltration membranes is presented in Table 2. Membranes that contain 2 wt. % TiO_2_ have higher surface porosity, which ultimately leads to superior membrane permeation properties [71]. The addition of TiO_2_ NPs, results in the improved interaction between NPs in the polymeric solution which have a positive effect on salt rejection performance compared to the pristine membrane. After the dispersion of NPs, significant effects on hydrophilicity and porosity were observed, and as a result, hydrophilicity was improved by up to 21%, and porosity increased by up to 76%. It was predicted that an improvement in hydrophilicity may lead to an improvement in their fouling performance. The effective number and the size of active pores depends on the process conditions, i.e., higher pressure opens more voids in the material of the polymer membrane and turns them into active pores. CA possesses the ability to remove NaCl. It was also observed that these salt rejections were higher than those reported in the literature in case of phase inversion method and for pristine NF membranes. As a result, small pore size and the interspacing of TiO_2_, which enhances the size exclusion.

Table 3 shows numerous research reports suggesting that NPs affect water flux and rejection performance. Wang and coworkers prepared NF membranes by using the interfacial polymerization of a layer of polyamide on a carbon nanotube and PES composite with ZIF–8 NPs. The results of flux (53.51 L/m^2^h) and salt rejection (95%) demonstrated higher performance for desalination [37]. Reza and coworkers investigated cellulose acetate/TiO_2_ hybrid membranes via the phase inversion method. The results showed that the CA–TiO_2_ membranes became more porous because of the increasing mean pore size. It is also shown that the addition of TiO_2_ NPs leads to increased water flux up to 47.42 L/m^2^h [64]. Lee and coworkers investigated novel polyamide nanocomposite membrane via the in-situ interfacial polymerization method. TiO_2_ NPs dispersed in the aqueous phase of phenyl diamine and organic phase (trimesoyl chloride). The results showed that salt rejection of up to 95% and permeation flux of 9.1 L/m^2^h [13]. Pourjafar and coworkers investigated novel PES/PVA nanofiltration membranes via dip coating in the presence of TiO_2_ NPs and a Glutaraldehyde (GA) cross-linker. The results showed a NaCl rejection up to 41% and water flux up to 44 L/m^2^h [68]. In a recent report, polymeric nanocomposite membranes prepared via the phase inversion process and PES/CA base polymers were loaded with titanium dioxide nanoparticles and PVP pore former, indicating a water flux of 89.6 L/m^2^h and salt rejection of 76%, along with an enhancement of other properties such as contact angle, porosity, and thermal stability.

## 4. Conclusions

The novelty of this study was lies with the synthesis of mixed-matrix nanocomposite membranes via the phase inversion process by dispersing TiO_2_ nanoparticles/PVP pore former in a CA/PES casting solution. In the first step, to optimize concentration of PVP, different concentrations (0, 2.5, 5, 7.5 and 10 wt. %) were used to improve the hydrophilicity, salt rejection, and flux of the membranes. Finally, 7.5 wt. % PVP was selected as the best concentration on the basis of the salt rejection results for a feed concentration of 1000 ppm NaCl solution. In the second step, different concentrations of TiO_2_ (0.5, 1, 1.5, and 2 wt. %) were used to investigate salt rejection and flux. The addition of TiO_2_ with pore former to the polymer casting solution had a significant impact on the membrane performance. FTIR analysis confirmed the presence of interaction between polymers and filler. SEM analyses confirmed a dense and smooth structure along with a uniform distribution of filler. TGA analyses proved that the thermal stability of MMNMs increased with an increase in filler loading. Energy required to break down the polymer chain increased, and the polymer chain rigidity was also increased. Contact angle analysis confirmed that the MMNMs were more hydrophilic in nature as the loading of TiO_2_ increasedinthe presence of an optimized PVP pore former concentration. Finally, the membranes were tested for desalination performance by measuring the water flux and salt rejection at various concentrations ranging from 1000 to 4000 ppm NaCl solution at 6 bars. It was found that the membrane with 2 wt. % TiO_2_ exhibited the best salt rejection of 79% and water flux of 89.6 L/m^2^h. Furthermore, TiO_2_ concentration increases resulted in the agglomeration of NPs, which subsequently decreased the permeation flux. These results prove that prepared membranes can be useful candidates for salt removal. It is interesting to note that increasing the TiO_2_ concentration seemed to have a lesser effect on membrane salt rejection. In general, the performance results showed that the PVP pore former played an important role in changing the structural properties of the membranes.

## Figures and Tables

**Figure 1 membranes-11-00433-f001:**
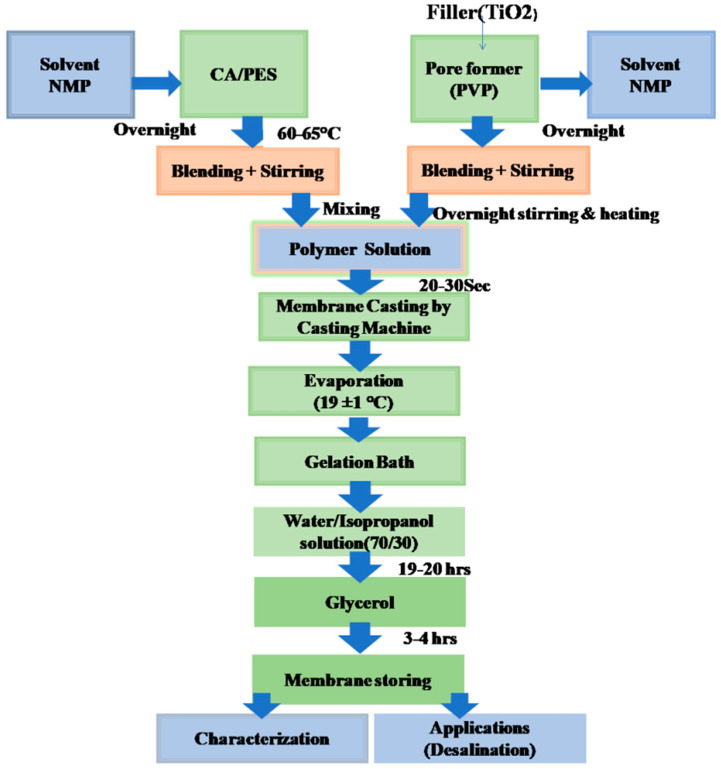
Preparation method of the mixed-matrix nanocomposite membranes.

**Figure 2 membranes-11-00433-f002:**
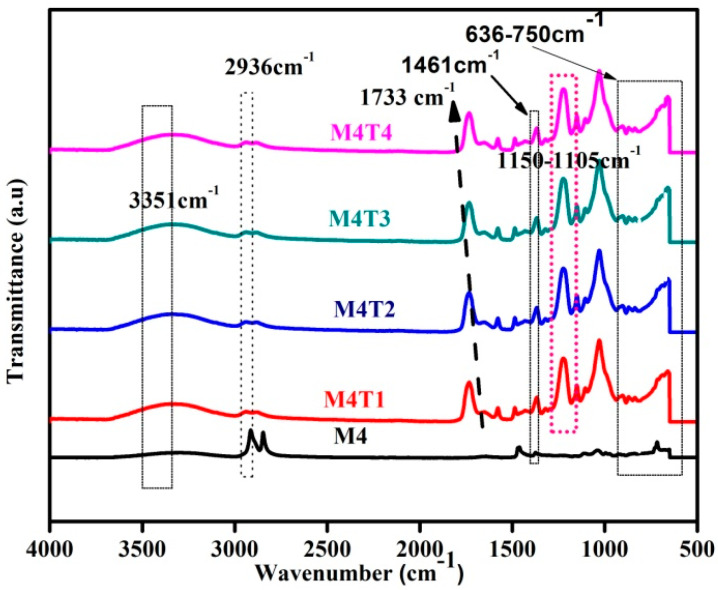
FTIR spectrum of the fabricated nanocomposite membranes.

**Figure 3 membranes-11-00433-f003:**
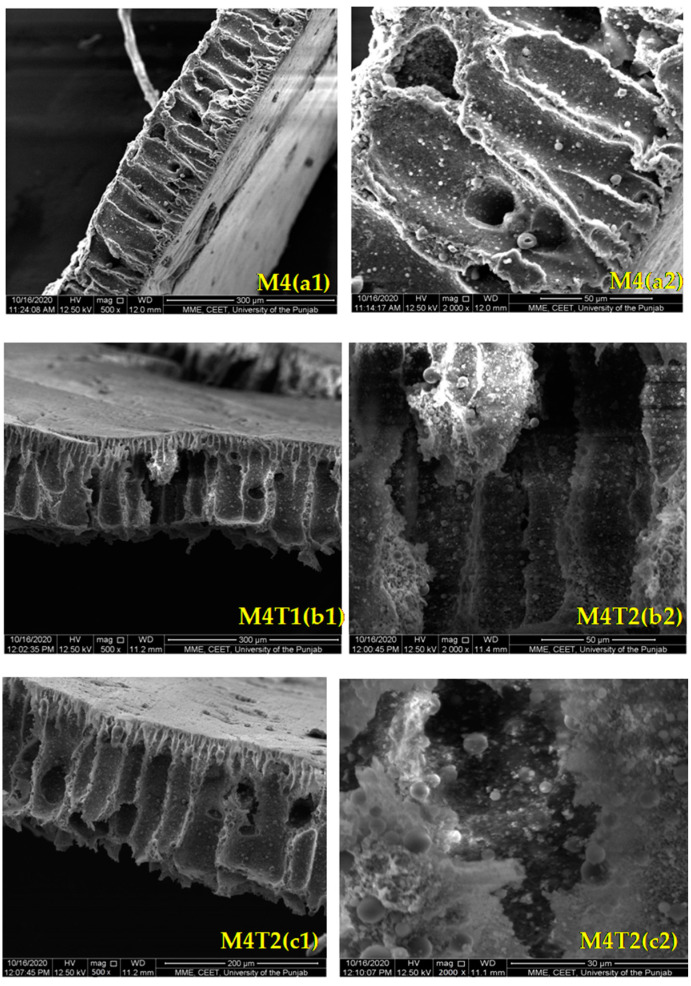
Cross-sectional SEM images of M4, M4T1, M4T2, M4T3, and M4T4.

**Figure 4 membranes-11-00433-f004:**
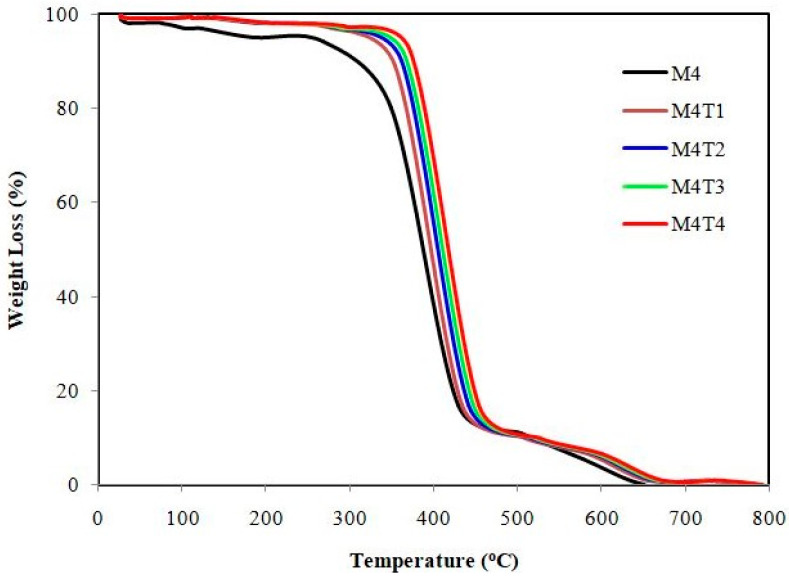
Thermograms of fabricated membranes.

**Figure 5 membranes-11-00433-f005:**
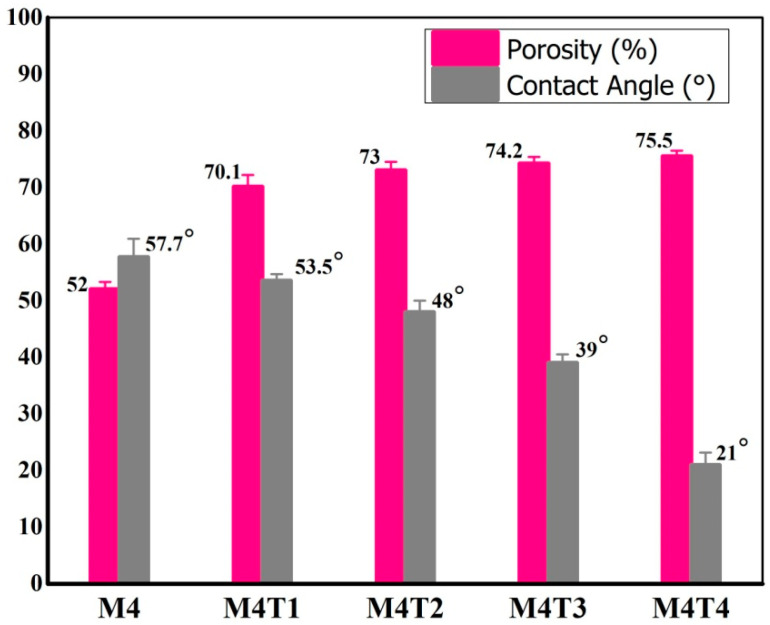
Pristine and nanocomposite membranes’ static water contact angle (average contact angles of samples are reported) and percent porosity.

**Figure 6 membranes-11-00433-f006:**
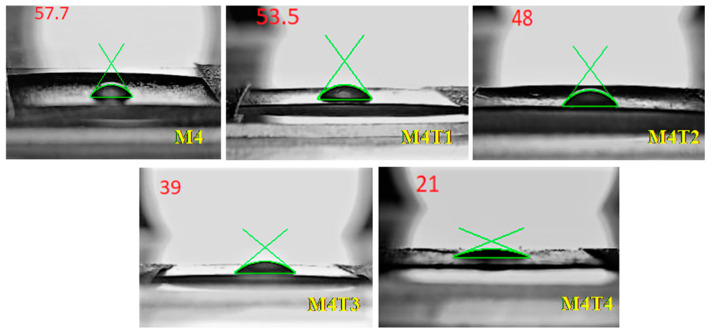
Static water contact angles: M4, M4T1, M4T2, M4T3, and M4T4.

**Figure 7 membranes-11-00433-f007:**
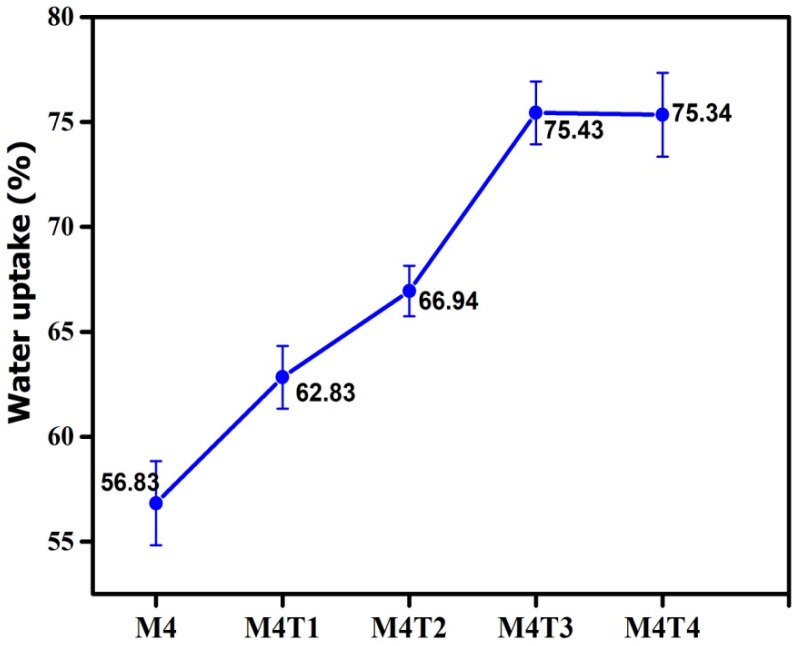
Effect of water uptake on MMNMs and the pristine membrane.

**Figure 8 membranes-11-00433-f008:**
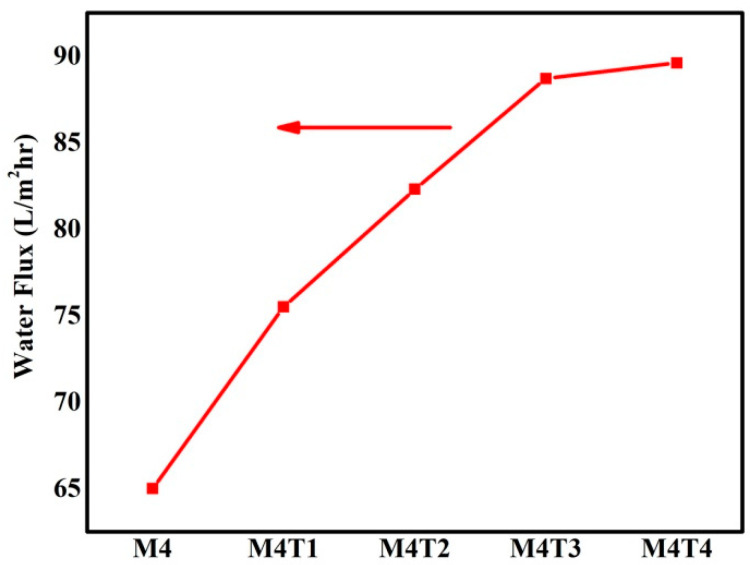
Water flux measurements of the prepared membrane with or without TiO_2_/pore former.

**Figure 9 membranes-11-00433-f009:**
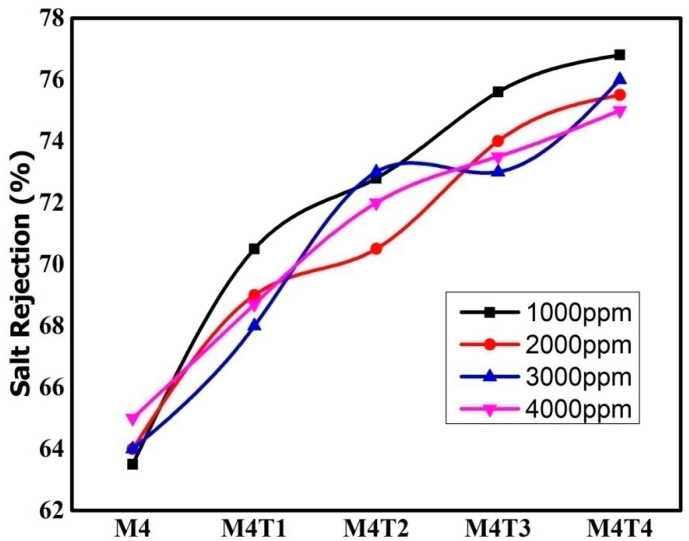
Salt rejection of the NaCl solution at various concentrations.

**Table 1 membranes-11-00433-t001:** Composition of the prepared membranes.

Membrane Type	Polymer		Pore Former	Particles
	CA (wt. %)	PES (wt. %)	PVP (wt. %)	TiO_2_ (wt. %)
M1	80	20	0	0
M2	78	19.5	2.5	0
M3	76	19	5	0
M4	74	18.5	7.5	0
M5	72	18	10	0
M4T1	74	18.5	7.5	0.5
M4T2	74	18.5	7.5	1
M4T3	74	18.5	7.5	1.5
M4T4	74	18.5	7.5	2

**Table 2 membranes-11-00433-t002:** TiO_2_/pore former effect on the performance of the pristine membrane and MMNMs.

Membrane Code	Porosity	Contact Angle	Water Uptake	Water Flux	Salt Rejection			
	(%)	(°)	(%)	L/m^2^h	(%)			
					1000 ppm	2000 ppm	3000 ppm	4000 ppm
M1	0	0	0	55	70.5	0	0	0
M2	0	0	0	60	68.1	0	0	0
M3	0	0	0	62.5	65.2	0	0	0
M4	52 ± 1.3	57.7 ± 3.2	56.83 ±2	65	63.5	64	64	65
M5	0	0	0	69.5	59.4	0	0	0
M4T1	70.1 ± 2.1	53.5 ± 1.2	62.83 ±1.5	75.5	70.5	69	68	68.7
M4T2	73 ± 1.5	48.0 ± 2.0	66.94 ±1.2	82.3	72.8	70.5	73	72
M4T3	74.2 ± 1.2	39.0 ± 1.5	75.43 ±1.5	88.7	75.6	74	73	73.5
M4T4	75.5 ± 1	21 ± 2.2	75.34 ±2	89.6	76.8	75.5	76	75

**Table 3 membranes-11-00433-t003:** Comparison between previous studies and the current study.

Polymer/Composite	NPs	Performance (Flux L/m^2^h)/ Salt Rejection%)		Ref.
Polyethersulfone, cellulose acetate, and Polyvinylpyrrolidone(PES/CA/PVP)	TiO_2_	89.6 L/m^2^h	76.8 ± 1	Present
Polyethersulfone and CNT composite	ZIF–8	53.51 L/m^2^h	95%	[37]
Cellulose acetate	TiO_2_	47.42 L/m^2^h	-	[64]
Polyamide nanocomposite	TiO_2_	9.1 L/m^2^h	95 %	[13]
Polyvinyl alcohol/PES	TiO_2_	44 L/m^2^h	41%	[68]

## Data Availability

Not Applicable.

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
