# Peer review of "TiO2 Nanoparticle Filler-Based Mixed-Matrix PES/CA Nanofiltration Membranes for Enhanced Desalination"

_membranes, 2021, doi:10.3390/membranes11060433_

Round 1
Reviewer 1 Report
The manuscript submitted by authors related to “TiO2 Nanoparticles Filler Based Mixed Matrix PES/CA Nanofiltration Membranes for Enhanced Desalination” is a good effort made by authors. Still manuscript needs major revision for reconsideration.
- Provide the reason for choosing Polyether sulfone and cellulose acetate as the principal polymer for membrane backbone. What is the major revision for selecting these two polymers.
- On page 2, Nanofiltration is one of the important techniques for making improvements in membrane performance for blending inorganic nanoparticles. Provide the relevant reference in support of this sentence.
- What is the reason for selection of TiO2 nanoparticle in present case. A clear justification need to be added under introduction section.
- Section heading 2.1. CA/PES/TiO2 Synthesis is not informative. Provide the brief but informative heading.
- Authors optimize the protocol used for the synthesis of CA/PES/TiO2. If so kindly, provide the impact of different concentrations of PES, CA and TiO2 NP for the fabrication of mixed matrix nano composite membranes
- Many place in the paper subscript and superscript is not provided properly. Kindly edit it throughout the script.
- Under section 2.2. MMNM Characterization, provide with the made of instruments used in the present work.
- Section 3.2. Scanning Electron Microscopy Study need more clear and deep discussion with proper justification.
- On page 8, under section 3.3. Thermo gravimetric Analysis, I cannot find any significant change in the thermal degradation of M4T1, M4T2, M4T3 and M4T4. Therefore, what information authors like to present in this section is not so clear.
- Conclusions section need to be shorten.
- Please write your text in good English (American or British usage is accepted, but not a mixture of these). English language manuscript require editing to eliminate possible grammatical or spelling errors and to conform to correct scientific English.
Author Response
Thanks for your positive comments for improving our manuscript. We have tried our best to improve the grammatical and spelling mistakes in the manuscript.

Reviewer 2 Report
The manuscript presented by Mehwish Batool and co-authors about TiO2 Nanoparticles Filler Based Mixed Matrix PES/CA Nanofiltration Membranes for Enhanced Desalination, the method and manuscript are interesting, the concentration ranging between 0 to 2 wt % of titanium dioxide nanoparticles (TiO2 NPs) was used for the study, and it was found that the membrane having 2 wt. % TiO2 exhibited the best results, and how is the results of higher concentration TiO2 NPs ? the authors should add more contrast experiments. The figures could be made better, e.g., the content of Figure 1 is not so clear for readers. Both of lower and higher magnification of SEM images (Figure 3) could be used in the manuscript, i.e., lower magnification for full images, and higher magnification for chosen small area. Some sentences were written not so well, the English should be improved, and there are some small mistakes could be found in the content and references, the mistakes should be corrected.
Author Response
Thanks for your comments that help us to improve the contents of manuscript and make the manuscript according to the journal requirements.

Reviewer 3 Report
Authors report about the desalination process via nanofiltration using polymer-based membranes enriched with TiO2 nanoparticles. The topic is well discussed and presented. A deep revision of the languange, punctuation and acronyms (i.e. in the introduction, RO must be explained) is need.
Author Response
We have thoroughly improved the manuscript and eliminate the grammatical and spelling mistakes.

Round 2
Reviewer 1 Report
The authors have satisfactory revised the manuscript.
Reviewer 2 Report
The authors had made the revision based on the comments. The present manuscript looks much better after the modification, and thus this it is recommended for publication.